# Biocidal Properties of New Silver Nanoparticles Argirium SUNc^®^ Against Food Hygiene Indicator Microorganisms

**DOI:** 10.3390/nano15040295

**Published:** 2025-02-14

**Authors:** Andrea Mancusi, Marica Egidio, Yolande Thérèse Rose Proroga, Luca Scotti, Hans Peter Deigner, Orlandina Di Maro, Santa Girardi, Marika Di Paolo, Raffaele Marrone

**Affiliations:** 1Department of Food Safety Coordination, Istituto Zooprofilattico Sperimentale del Mezzogiorno, 80055 Portici, Italy; andrea.mancusi@izsmportici.it (A.M.); yolande.proroga@izsmportici.it (Y.T.R.P.); orlandina.dimaro@izsmportici.it (O.D.M.); santa.girardi@izsmportici.it (S.G.); 2Department of Veterinary Medicine and Animal Production, University of Naples Federico II, 80138 Naples, Italy; marica.egidio@libero.it (M.E.); marika.dipaolo@unina.it (M.D.P.); raffaele.marrone@unina.it (R.M.); 3IIS “Alessandrini-Marino” Via Marino, 64100 Teramo, Italy; 4Institute of Precision Medicine, Furtwangen University, Jakob-Kienzle Str. 17, 78054 Villingen-Schwenningen, Germany; deigner@gmx.de

**Keywords:** biocides, resistance, Argirium SUNc^®^, meat cutting and processing company, seafood processing plant, dairy plant

## Abstract

Microbial resistance to conventional biocides is closely linked to the more complex problem of antibiotic resistance. Therefore, the development of novel and highly antimicrobial effective disinfectants is encouraged. Due to their broad spectrum of action and low toxicity, Argirium Silver Ultra Nano Clusters (Argirium SUNc^®^), a new generation of silver nanoparticles, could be one of them. In this regard, the aim of the present work was to evaluate their biocidal properties in two different formulations against the hygiene indicator microorganisms potentially present in three different Italian food industries and to compare them with the chemical disinfectant most commonly used by operators for routine cleaning. Therefore, a series of microbiological swabs on different foodstuff contact surfaces were performed before and after the application of the solutions at each food company. The data showed that this novel nanomaterial was effective against all the parameters analyzed, being able to inhibit or reduce the growth of the tested microorganisms. Furthermore, in most cases, the two sanitizing solutions tested had a greater inhibitory power than the conventional disinfectant. For this reason, Argirium SUNc^®^ has great potential to be used in the near future as a new-generation disinfectant, an alternative to conventional disinfectants that promote the spread of antibiotic resistance.

## 1. Introduction

Antibiotic resistance, due to the ability of bacteria to survive and proliferate even in the presence of an antibacterial agent, has spread worldwide, posing a serious threat to public health. The use of the same chemical products in multiple sectors, such as the medical, pharmaceutical, chemical, and food industries, is responsible for this phenomenon, bringing about the need to search for new agents with high antimicrobial activity but with different mechanisms of action from those of available antibiotics, to which resistance is neither established nor widespread. For this reason, several studies are being carried out to develop innovative antimicrobial products and to test their properties in vitro and in vivo. Nanotechnologies give hope in this direction, opening new horizons to combat the resistance of microorganisms to drugs [1]. In fact, it has been observed that the antimicrobial activity of these materials against various multidrug-resistant Gram-positive and Gram-negative microorganisms has significant positive correlations related to dose-dependent effects [2]. For this reason and because of their unique characteristics and capabilities, metal-containing nanoparticles such as silver, copper, titanium, zinc, and gold are increasingly being incorporated in a wide range of consumer products as antimicrobial materials [3].

### 1.1. Silver Nanoparticles (AgNPs)

Silver nanoparticles (AgNPs), for their physical (high surface-area-to-volume ratio), chemical, and biological properties [4,5,6,7], are the most studied and utilized [8], as also observed in the previous work [9]. Numerous studies have demonstrated their broad spectrum of activity against either Gram-positive or Gram-negative bacteria [10], some fungi, and some viruses [11,12] and their low propensity to induce microbial resistance [13]. AgNPs’ bactericidal power against many microorganisms has been investigated by various teams: in a study on the antibacterial properties of nanoscale gold, silver, and zinc oxide against *Streptococcus mutans*, Hernández-Sierra et al. found that AgNPs were the most effective [14]. Sondi and Salopek-Sondi observed that AgNPs were also effective against *Escherichia coli,* causing the formation of some “pits” in the bacterial cell wall with consequent accumulation in the cellular membrane, augmented permeability of the cell wall, and finally cell death [15]. AgNPs were able to kill or inhibit *Staphylococcus aureus* and *Salmonella typhimurium*, as shown by Shameli et al., and their antibacterial activity strongly relied on the dimensions of the particles [16], because a smaller nanoparticle penetrates more easily into the bacterial cell membrane and dissolves faster, releasing more Ag^+^ [17]. With regard to multidrug-resistant *Pseudomonas aeruginosa*, Shijing Liao et al. found that AgNPs exhibited effective bactericidal activity, causing bacterial death through a disequilibrium of oxidation and antioxidation processes with the failure to eliminate the excessive reactive oxygen species (ROS) [18]. Furthermore, it has been shown that silver is also particularly effective against bacteria due to its ability to dissolve and ionize in solutions such as water, body fluids, and organic tissues [3,19]. This aspect makes it a safe inorganic antibacterial agent that kills many types of pathogenic microorganisms.

Due to its antimicrobial properties, interest in colloidal silver has grown in the last decade, and AgNPs are promising for use as a potent antibacterial agent against multidrug-resistant microorganisms. Therefore, more and more products containing silver nanoparticles are expected to enter the market in the next decade [20]. Many researchers are also working to create surfaces with antimicrobial properties [21]. In this regard, Pazos-Ortiz et al. designed a material of silver nanoparticles (AgNPs) embedded in polyepsilon–caprolactone nanofibers [22].

Currently, the best known AgNP antimicrobial mechanisms of action include the following:Disruption of bacterial cell wall integrity with a consequent increase in membrane permeability and leakage of the cell constituents, with eventual cell death;Interruption of the respiratory chain reaction (bacterial respiration and ATP synthesis) by combining sulfhydryl with oxidative damage of DNA and proteins and subsequent cell death [23,24];Binding to the phosphorous groups of DNA, disrupting its transcription and translation [25];Dephosphorylation of phosphotyrosines, with alteration of the cell signal transduction process and cell death;The interaction of the Ag^+^ ions released from the nanoparticles interacting with the cell membrane and cell wall components of the bacteria, which is one of the crucial mechanisms of toxicity of AgNPs [26].

However, some antimicrobial mechanisms of action of AgNPs are still unknown. Furthermore, important challenges need to be solved to make their synthesis methods reproducible, because sometimes the results expected from nanoparticle formulations are altered by the presence of reducing agents, stabilizers, and contaminants that could interfere with the efficacy and toxicity of AgNPs [27,28].

### 1.2. Argirium Silver Ultra Nano Clusters (Argirium SUNc^®^)

For this reason, novel silver nanoparticles were created using a reproducible electrochemical method (Patent EP-18181873) derived from the modification of an old synthesis method [29]. These nanoparticles, called Argirium Silver Ultra Nano Clusters (Argirium SUNc^®^), are the smallest of all nanoparticles studied so far, with a size < 2 nm [30]. They have a particle structure never before observed in a stable form, with a core of metallic Ag^0^ and external shells made by Ag^+^, Ag^2+^, and Ag^3+^ silver oxides. These characteristics are responsible for their unique chemo-physical properties. In fact, Argirium SUNc^®^ nanoparticles are stable for several months in ultra-pure water solutions thanks to a high anionic solvation (Zpuls value > −50 mV) [31] and have enhanced redox properties towards biological targets due to the presence of silver oxides on the cluster surface [30]. Therefore, they are considered a novel nanomaterial that has shown antimicrobial activity against either susceptible or resistant strains at a concentration much lower than that reported for other silver formulations (<1 µg/mL) [32]. Furthermore, Argirium SUNc^®^ has been shown to act on bacterial biofilm structures, resulting in their disassembly (0.625 µg/mL) [33,34,35]. Regarding their toxicity, many studies in the literature indicate that small nanoparticles have a higher surface area/volume ratio and a stronger bond with the bacterial cell membrane than larger ones responsible for a greater cytotoxicity [36]. However, recent in vitro and in vivo toxicity studies carried out on the Argirium SUNc^®^ nanoparticles against lung epithelial cells revealed that they are not toxic up to a concentration equal to 100 µg/mL [37]. Another study has shown that Argirium SUNc^®^ was about 10 times less toxic to human cells than to bacteria [34]. In addition, the same author teams tested them with *Galleria mellonella* as a pre-clinical toxicity test [34]. The survival curves of the larvae showed that Argirium SUNc^®^ nanoparticles were non-toxic up to the highest concentration possible used: 6.8 µg/mL [34]. For this reason, this new type of silver nanoparticle could represent a turning point for the future. Specifically, since resistance to conventional biocides used in the food industry could be involved in the bigger problem of antibiotic resistance, Argirium SUNc^®^ nanoparticles could be used as an antimicrobial alternative and innovative principle to produce a new generation of bio-gels compatible in the medical field and of preparations for environments and equipment sanitization which do not promote resistance.

In this regard, the aim of the present work was to evaluate the biocidal properties of this novel nanomaterial used as a disinfectant in different formulations (spray and drop) and concentrations (1.5, 2, 3, and 4 µg/mL) against the hygiene indicator microorganisms potentially present in a seafood processing plant, a dairy plant, and a meat cutting and processing plant and to compare their effectiveness with that of the disinfectant most used by operators in the food industry of interest.

## 2. Results

The results of the comparison of the antimicrobial performance of the three types of disinfectants (a conventional chlorine-based disinfectant and Argirium SUNc^®^ in spray and in drop formulation) on the hygiene indicator microorganisms (total bacterial count, enterobacteria, coagulase-positive staphylococci, *Escherichia coli*, yeasts, and molds) found on the surfaces (three environments, three tools, and three equipment types for each food industry) selected in the three food companies studied (a seafood processing plant, a dairy plant, and a meat cutting and processing plant) are shown in Figure 1, Figure 2 and Figure 3. The graphs show an average reduction percentage (%) calculated for each parameter analyzed after a 10 min exposure to each disinfectant. Specifically, the summary data presented in Figure 1 compare the antimicrobial efficacy of the three sanitizer solutions when applied to environmental surfaces across the three food companies. Similarly, Figure 2 shows a comparison of their biocidal effects against the microorganisms isolated from tools, while Figure 3 compares their biocidal efficacy against microorganisms isolated from equipment.

### 2.1. Environmental Surfaces

In summary, the data presented in Figure 1 show the high biocidal activity of both Argirium SUNc^®^ formulations against the hygiene indicator microorganisms isolated from the environmental surfaces chosen across the three examined food companies, with a notable reduction in the number of viable bacteria compared to the untreated control and their greater antimicrobial effectiveness compared to the disinfectant currently used in the target food industries against all the parameters analyzed.

In particular, with respect to the total bacterial count (TAB 30 °C), the data showed that Argirium SUNc^®^ in spray formulation exhibited the highest efficacy against the TAB at 30 °C of the meat cutting and processing company, while the drop formulation demonstrated superior effectiveness in the seafood processing plant. Only in the dairy plant was the conventional disinfectant found to be more effective than the innovative sanitizers.

The situation was different for enterobacteria (Ent). In this case, both the Argirium SUNc^®^ formulations in the meat cutting and processing company and the drop form in the other two food companies exhibited higher biocidal activity compared to the conventional disinfectant and proved to be more effective in inhibiting the growth of enterobacteria, with a reduction of almost 100%.

Regarding coagulase-positive staphylococci (C-staf+), where comparisons were possible (some surfaces could not be assessed due to negligible microbial load), a higher reduction (%) in microbial proliferation was observed in the seafood processing plant after applying the innovative sanitizer solutions, indicating their superior efficacy compared to the conventional disinfectant. Greater effectiveness was also observed for the spray formulation in the dairy farm, although comparisons with the conventional disinfectant were not possible due to the negligible microbial load present on these specific areas of the selected surfaces. An exception was observed for coagulase-positive staphylococci isolated from the environmental surfaces of the meat cutting and processing company, where the conventional disinfectant proved to be more effective than the Argirium SUNc^®^ formulations. Additionally, it is important to highlight that among the coagulase-positive staphylococci, two important microorganisms were isolated (and identified) from the targeted food companies. They were as follows:

*Staphylococcus aureus*, a known foodborne pathogen, was found on outgoing tape of the meat cutting and processing company and on the outer surface of a plastic bucket used for wastewater and a forming tube of the dairy plant;

*Shewanella xiamenensis*, an emerging zoonotic pathogen [38], was found on a plastic basket containing seafood, a plastic ladle for bivalve mollusks sorting, a marble cutting board, and an automatic clipping machine of the seafood processing plant.

The data obtained showed that the Argirium SUNc^®^ nanoparticles used as sanitizing solutions were also able to act on these bacteria, notably reducing their growth. Our findings on *Staphylococcus aureus* are consistent with previous studies demonstrating the antimicrobial effectiveness of Argirium SUNc^®^ against the same microorganism [9,35], while for *Shewanella xiamenensis*, to the best of our knowledge, these are the first results on the biocidal properties of Argirium SUNc^®^ against this microorganism, making comparisons with other studies currently impossible.

Comparing the antimicrobial effectiveness of the three biocides on yeasts, the conventional disinfectant was found to be more effective in the meat cutting and processing company, while Argirium SUNc^®^ in drop formulation showed a greater biocidal activity in the dairy plant. Data for the spray formulation in the dairy plant are unavailable, as the microbial load on the chosen area of the selected surfaces was negligible. A similar situation was observed in the seafood processing plant, where both Argirium SUNc^®^ formulations showed the same high antimicrobial effectiveness (100%). However, a comparison with the conventional disinfectant was not possible due to the absence of microbial growth on the agar plates.

Finally, regarding molds, data are only available for the meat cutting and processing company, where the spray formulation proved to be the most effective disinfectant, resulting in a 100% inhibition. No data were recorded from the other food companies, as no mold growth was detected on the environmental surfaces tested.

### 2.2. Tool Surfaces

The summary data presented in graph 2 (Figure 2) indicate the distinct antimicrobial performance of both Argirium SUNc^®^ formulations in comparison to the conventional disinfectant after being applied to the selected tool surfaces. The findings showed notable variations in the biocidal activity between the different disinfectants. However, contrary to the results obtained for environmental surfaces, in this case, the conventional disinfectant demonstrated greater effectiveness than the innovative biocidal solutions in reducing or inhibiting the growth of hygiene indicator microorganisms, particularly enterobacteria, coagulase-positive staphylococci, and molds isolated from the surfaces of the selected tools across the three types of companies, although substantial reduction percentages were also observed for both Argirium SUNc^®^ formulations (Figure 2).

Notably, Argirium SUNc^®^ in the spray formulation exhibited superior antimicrobial effectiveness against TAB 30 °C isolated from the tool surfaces in both the meat cutting and processing company and the seafood processing plant. In fact, this formulation resulted in a greater reduction in microbial growth compared to the conventional disinfectant (Figure 2), which showed higher biocidal power only within the dairy plant.

The Argirium SUNc^®^ spray formulation was found to be the most effective in reducing enterobacteria growth on the tool surfaces of the meat cutting and processing company, while the conventional disinfectant displayed higher biocidal activity against enterobacteria present in the seafood and dairy plants.

The antimicrobial activity observed on coagulase-positive staphylococci was also noteworthy. In the seafood processing plant, the Argirium SUNc^®^ in spray formulation resulted in an almost total inhibition (about 99%) of bacterial proliferation on tools, outperforming the conventional disinfectant. However, in both the dairy plant and the meat cutting and processing company, the conventional disinfectant demonstrated superior efficacy compared to both Argirium SUNc^®^ formulations in reducing staphylococcal contamination on tools, although with a minimal difference in percentage reduction in the dairy plant (reduction of 99% observed with the conventional disinfectant compared to a 98% reduction for the spray form).

Regarding yeasts, the conventional disinfectant was found to be most effective in the dairy plant, resulting in a higher biocidal activity compared to Argirium SUNc^®^. Conversely, in the other two food companies, both Argirium SUNc^®^ formulations outperformed the conventional disinfectant in reducing yeast growth on the tool surfaces tested. Notably, in the meat cutting and processing company, the drop formulation of Argirium SUNc^®^ exhibited superior antimicrobial activity, while in the seafood processing plant, the spray formulation was found to be more effective, although comparisons with the drop formulation were not possible due to the negligible microbial growth on the selected area of the tested tool surfaces.

Finally, regarding molds, the conventional disinfectant exhibited the highest efficacy in controlling the mold populations of the dairy and seafood plants (100% inhibition), whereas Argirium SUNc^®^ in drop formulation showed greater biocidal activity in the meat cutting and processing company, achieving a total inhibition (100%) of microbial growth compared to a 50% inhibition rate observed with the conventional disinfectant.

### 2.3. Equipment Surfaces

Graph 3 (Figure 3) focuses on the antimicrobial effectiveness shown by the three biocides on equipment surfaces selected in the three different food companies. In summary, where comparisons were possible, a distinctly different pattern emerged compared to the other two types of surfaces previously analyzed. In fact, as observed with the environmental surfaces, the overall data related to the total bacterial count and enterobacteria further highlighted the greater inhibitory potential of both Argirium SUNc^®^ formulations compared to the chlorine-based disinfectant currently used in the food companies of interest and their potent biocidal activity. However, in contrast to previous observations, on the equipment surfaces, both conventional and innovative sanitizer solutions exhibited comparable effectiveness in controlling coagulase-positive staphylococci, yeasts, and molds.

Notably, regarding the TAB at 30 °C, the Argirium SUNc^®^ spray formulation exhibited the highest antimicrobial performance, showing a higher reduction in microbial proliferation compared to both the Argirium SUNc^®^ drop formulation and the conventional disinfectant (Figure 3) in both the meat cutting and processing company and the dairy plant. In contrast, the conventional disinfectant performed better against the TAB at 30 °C at the seafood processing plant.

A different trend was observed for enterobacteria, with both Argirium SUNc^®^ formulations outperforming the conventional disinfectant across all the three food companies by effectively inhibiting microbial growth, achieving up to 100% inhibition in the seafood processing plant. Notably, the Argirium SUNc^®^ drop formulation emerged as the most effective disinfectant in both the meat and dairy plants, while in the seafood processing plant, the two innovative sanitizing solutions (spray and drop) exhibited equally high biocidal activity on enterobacterial growth (Figure 3). This result is particularly significant, as enterobacteria serve as hygiene indicators and are associated with potential fecal contamination. Finally, it is noteworthy that the percentage reductions in enterobacteria growth following the application of the biocides showed minimal variation between the three disinfectants and across all three companies, as illustrated in Figure 3.

Regarding coagulase-positive staphylococci (C-staf+), the Argirium SUNc^®^ formulations, particularly the drop formulation, were notably effective in the meat cutting and processing company, resulting in a higher microbial reduction compared to the conventional disinfectant (Figure 3). In the seafood processing plant, the drop formulation had the same effectiveness of the conventional disinfectant in inhibiting C-staf+ growth (100%), while in the dairy plant, the latter showed superior performance compared to the Argirium SUNc^®^ formulations in controlling this microbial parameter.

For yeasts and molds, there are no data related to the seafood processing plant because of the negligible microbial load found the selected equipment surfaces.

At the meat processing plant, the Argirium SUNc^®^ disinfectant in drop formulation demonstrated slightly higher efficacy against yeasts compared to both the spray form and the conventional disinfectant. Conversely, for molds, the spray form of Argirium SUNc^®^ exhibited a marginally higher effectiveness, with minimal differences in percentage reduction observed (Figure 3).

A different situation was observed in the dairy plant, where the conventional disinfectant displayed the highest efficacy in reducing both yeast and mold populations, showing superior performance in microbial control compared to the other disinfectants tested (Figure 3).

Finally, with regard to *E. coli*, it is important to underline that this microorganism was only isolated on some food contact surfaces of the meat cutting and processing company. Where it was found, both innovative Argirium SUNc^®^ formulations used for sanitization showed their antibacterial effectiveness by partially or totally eliminating it, and where the inhibitory properties of the biocides could be compared, both Argirium Sunc^®^ formulations demonstrated greater effectiveness than the conventional disinfectant.

## 3. Discussion

In the present work, two sanitizing solutions (one in drop formulation and another one in spray formulation) composed of a new type of silver nanoparticles (Argirium SUNc^®^) were tested at different concentrations against the hygiene indicator microorganisms possible present in three different Italian food industries (a seafood processing plant, a dairy plant, and a meat cutting and processing plant). The biocidal potential of these innovative solutions was then compared with that of the chlorine-based disinfectants used for routine cleaning in the food industries of interest.

According to other studies [9,31,35,39], the overall data have highlighted the broad spectrum of action of Argirium SUNc^®^ and its prominent antimicrobial properties at a concentration (1.5 µg/mL) much lower than that reported for other silver preparations [32]. In general, the antibacterial activity of AgNPs ranges between 10 and 100 µg/mL [32], while in the present study, the Argirium SUNc^®^ nanoparticles used as a biocide were able to partially or completely inhibit microbial proliferation in the range of 1.5 to 4 µg/mL. These properties, due to their unique antimicrobial mechanism responsible for their antibacterial and antifungal efficacy at a very low concentrations, are related to their particle structure (with Ag^2+^ and Ag^3+^ cationic forms stable in ultra-pure water solution), which distinguishes Argirium SUNc^®^ from any other nanomaterial [31]. In fact, the main biological target of Argirium SUNc^®^ is the cell membrane, whose depolarization with the subsequent increase in intracellular calcium levels and the loss of functions leads to bacterial and fungal death [31,35]. Another important property of Argirium SUNc^®^ (not reported in other silver formulations) is that the values of the Nerst equation lead to having a redox power greater than that of the oxygen–ozone couple, giving the product a degree of unique oxidative capacity [30]. Moreover, the same particle structure makes Argirium SUNc^®^´s formulation more effective than other commercial nanocomposites in destructuring mature biofilms, as it interferes with the cross-links necessary for its three-dimensional structure [30,35]. These characteristics were also observed by Molina-Hernandez et al. [35] against MDR strains, indicating that Argirium SUNc^®^ can overcome resistance and potentially be used to synthetize a new generation of disinfectants.

In fact, in addition to their greater effectiveness compared to other silver formulations as demonstrated in previous studies [31,32,35], Argirium SUNc^®^ nanoparticles were also shown in the present work to be more effective than the conventional chlorine-based disinfectants currently used in the considered food industries. The data obtained showed that both silver formulations had a greater ability to inhibit/reduce microbial growth compared to the conventional disinfectants on most surfaces of all three food companies.

Notably, the Argirium SUNc^®^ spray formulation was more effective than the drop formulation against the total bacterial count, yeasts, and enterobacteria on all tested surfaces (environmental surfaces, tools, and equipment).

For coagulase-positive staphylococci and molds, where comparisons were possible, the drop formulation was found to be more effective on tools and equipment, whereas the spray was found to be more effective on environmental surfaces.

Finally, both Argirium Sunc® formulations showed similar inhibitory properties against E. coli isolated only from some food contact surfaces of the meat cutting and processing company.

It is also important to underline that, as illustrated in Figure 1, Figure 2 and Figure 3, the efficacy demonstrated by Argirium SUNc^®^ does not appear to be influenced by the type of processed matrix (meat, fish, or milk) or by the concentration used. These findings suggest that, although Argirium SUNc^®^ solutions are effective across different environments, their performance may be impacted by the specific microbial profiles present in each facility.

In fact, the two innovative sanitizer solutions demonstrated varying levels of effectiveness, performing better on certain surfaces than others, because the activity of an antimicrobial agent depends on several factors, some of which are intrinsic properties of the organism and others that are derived from the chemicals and the external physical environment [40]. These need to be listed in more detail:Number and type of microorganisms: The meat cutting and processing company harbors a different microflora and initial level of microbial contamination compared to that found in the dairy plant or in the seafood processing plant. No one disinfectant can be effective against all classes of microorganisms [40].Surface type: Sanitizing complex instruments with multiple components or joints and channels, such as meat bagging machines or pasteurizers, is more difficult than sanitizing flat surfaces such as a work table. It is also important to consider the location of the microorganisms, as the presence of dirt can interact with the biocide, reducing its availability, or interact with the microorganisms, providing protection. Only surfaces in direct contact with the germicide can be sanitized [40].Material characteristics of a surface: This significantly impacts microorganism survival. For example, porous surfaces are more challenging to clean and, consequently, to disinfect, favoring microorganism survival. Similarly, wooden surfaces are more difficult to disinfect than steel surfaces [40].Precleaning process: Pretreatment of surfaces, especially when visibly soiled, is essential to ensure or improve the biocidal efficacy [40].

## 4. Materials and Methods

To evaluate the antimicrobial activity of Argirium SUNc^®^ nanoparticles used as a biocide against hygiene indicator microorganisms (coagulase-positive staphylococci, *Escherichia coli*, enterobacteria, total bacterial count, yeasts, and molds), three experimental tests were carried out in three different Italian food industries with different production processes. The Italian food industries were a seafood processing plant, a dairy plant, and a meat cutting and processing company.

### 4.1. Experimental Design

For each test, the experimental design envisaged the following:The comparison between the effectiveness of three different sanitizer solutions: two innovative biocides, one in spray formulation and another one in drop formulation made only with Argirium SUNc^®^ nanoparticles, and a conventional chlorine-based disinfectant commonly used in the food industry under consideration;The use of four different concentrations (1.5 µg/mL and 2 µg/mL for the meat cutting and processing company, 3 µg/mL for the dairy plant, and 4 µg/mL for the seafood processing plant) of the two innovative sanitizer solutions (spray and drop formulation) on each tested surface;The execution of 54 swabs (6 for each surface) before and after applying the solutions for a 10 min period on 9 different foodstuff contact surfaces (3 different tools, 3 different equipment and 3 different environments of the food industry of interest).

#### Argirium SUNc^®^ Nanoparticle Generation 

The tested silver ultra nanoclusters (Argirium SUNc^®^), provided by ARGIRIUM SUNc (srl Benefit corporation, Ostuni, Italy), were electrochemically synthesized by using an improved synthetic protocol in ultra-pure water without stabilizing agents or other chemical components. The synthesis method is protected by the European patent EP-18181873.

During the present experimentation, to produce the spray and drop formulations at the required Argirium SUNC^®^ concentrations ranging from 1.5 µg/mL to 4 µg/mL, dilutions were performed using sterile water. All Argirium SUNc^®^ concentrations used were below the toxic limit (6.8 µg/mL) [34].

### 4.2. Sample Collection

A total of 162 microbiological swabs (54 from each one) in three different Italian food industries were performed using cellulose sponge swabs (Whirlpak Cellulose Speci-Sponge Bags, Whirlpak, Fort Atkinson, WI, USA) moistened with 9 mL of sterile buffered peptone water (BPW, Buffered Peptone Water Dilucup, Thermo Fischer Scientific, Waltham, MA, USA).

Briefly, the moistened swab was wiped on the delimited area (10 × 10 cm) of the chosen surface at least ten times in both a vertical and horizontal direction. Then, all samples were sealed, identified, and placed on ice for transportation to the laboratory.

For each food industry, nine food contact surfaces (3 different environments, 3 different tools, and 3 different equipment types) were analyzed:A refrigerator cell’s wall, a stainless-steel meat contact trolley, and outgoing tape (as environments); a knife blade used to shred beef, a bone saw blade, and a meat bagger’s paddle (as tools); and a scale plate, a pig slaughter table, and a steel tub for meat bagging (as equipment) for the meat cutting and processing company;A steel worktable, a steel curd draining trolley, and an environmental wall (as environments); a curd break oar, the outer surface of a plastic bucket used for wastewater, and a cheese vat (as tools); and a forming tube for mozzarella cheese, a weighing scale, and a conveyor belt (as equipment) for the dairy plant;A refrigerator cell’s wall, a plastic basket containing seafood, and a steel sink (as environments); a fish filleting blade, a plastic ladle for bivalve mollusk sorting, and a marble cutting board (as tools); a scale for bivalve mollusk weight, an automatic clipping machine, and an automatic weighing/packaging machine (as equipment) for the seafood processing plant.

On all the chosen surfaces, six different swabs were performed:A first swab on a part of the selected surface before using any sanitizing solution.A second swab on another area of the same surface after having applied the Argirium SUNc^®^ sanitizer in the spray formulation at a concentration of 2 µg/mL on the food contact surfaces of the meat cutting and processing company; 3 µg/mL on the food contact surfaces of the dairy plant; and 4 µg/mL on the food contact surfaces of the seafood processing plant. After spraying, the sanitizing solution was distributed evenly and was left in contact with the surface for 10 min before performing the swab.A third swab on a third area of the selected surface before using any sanitizing solution.A fourth swab on a fourth area of the same surface after using the Argirium SUNc^®^ sanitizer in the drop formulation at a concentration of 1.5 µg/mL on the food contact surfaces of the meat cutting and processing company, 3 µg/mL on the food contact surfaces of the dairy plant, and 4 µg/mL on the food contact surfaces of the seafood processing plant, homogeneously distributed and left in contact with the surface for 10 min.A fifth swab on a fifth area of the selected surface, before using any sanitizing solution.A sixth swab on a sixth area of the same surface after the application of the disinfectant used for the routine cleaning of the food industry of interest, homogeneously distributed and left in contact with the surface for the same time period as chosen for the previous disinfectants.

### 4.3. Microbiological Evaluation

According to UNI EN ISO 18593: 2018 [41], all samples were transported to the Department of Food Safety Coordination, Istituto Zooprofilattico Sperimentale del Mezzogiorno, in a cool box at approximately 5 ± 3 °C. Then, microbiological analyses were carried out within 24 h [41] to test the antimicrobial effectiveness of Argirium SUNc^®^ nanoparticles against the following hygiene indicator microorganisms that may be present in the analyzed food industries: coagulase-positive staphylococci, *Escherichia coli*, enterobacteria, total bacterial count, yeasts, and molds.

#### 4.3.1. Microbial Identification with Selective Culture Media

In this regard, each cellulose sponge swab was weighed and diluted in 90 mL buffered peptone water (Dilucup, Thermo Fischer Scientific, Whaltman, MA, USA) such as to respect a 1:10 sample-to-solution ratio. Serial decimal dilutions were then prepared to determine coagulase-positive staphylococci, *Escherichia coli*, enterobacteria, total bacterial count, yeasts, and molds. The number of coagulase-positive staphylococci was determined on Baird-Parker agar (Biolife, Italy) incubated at 37 °C ± 1 °C for 48 h [ISO 6881-1:2021] [42]. The number of *E. coli* was counted on Tryptone Bile X-glucuronide (TBX) agar (Biolife, Italy) incubated at 44 °C ± 1 °C for 24 h ± 2 h, and enterobacteria were counted on Violet Red Bile Glucose Agar (VRBG, Biolife, Italy) through incubation at 37 °C ± 1 °C for 24 h ± 2 h [ISO 21528-1:2017] [43]. The number of yeasts and molds was determined on Dichloran Rose Bengal Chloramphenicol agar (DRBC, Biolife, Italy) through incubation at 25 °C ± 1 °C for 5/7 days [ISO 21527-1:2008] [44]. Finally, the total bacterial count was determined according to ISO 4833-1, using plate count agar (PCA, Biolife, Italy) through incubation at 30 °C ± 1 °C for 72 h ± 3 h [ISO 4833-1:2013] [45]. After incubation, colonies were counted to obtain data, expressed as CFU/cm^2^.

#### 4.3.2. MALDI-TOF MS Identification

For further identification of the microbial strains grown on agar plates, a MALDI-TOF MS (MALDI Biotyper^®^ Sirius, Arcore, Italy) analysis was carried out first using the “direct colony identification method” [46]. Briefly, colonies were smeared in duplicate onto a 96-pointsteel plate (Bruker Daltonics, Bremen, Germany). Then, samples were covered with a 1 µL matrix solution containing 10 mg/mL of α-cyano-4-hydroxycinnamic acid in acetonitrile (Sigma-Aldrich, Berlin, Germany), deionized water, and trifluoracetic acid (50:47.5:2.5, [*v*/*v*/*v*]). Bruker’s Bacterial Test Standard (BTS Bruker Daltonics) was used as the reference standard for the mass calibration and Flex Control 3.4 software (Bruker Daltonics, Bremen, Germany) was set in a linear positive ion detection mode (Bruker Daltonics). Isolates were analyzed by matching the collected spectra to those containing in the Bruker MSP database (MBT Compass Library) using the Bruker Compass software 2.0 at default settings. Identification score criteria were classified as follows: Samples were processed in a Microflex™ LT MALDI-TOF mass spectrometer (Bruker Daltonics) equipped with a nitrogen laser (l1⁄4337 nm) operating in linear positive ion detection mode using MALDI Biotyper Automation Control 2.0 (Bruker Daltonics). Identifications were obtained by comparing the mass spectra to the Bruker MSP database (version DB5989) using the Bruker Compass software (Bruker Daltonics) at default settings. Identification score criteria were classified according to Jeong et al. [47]: a score ≥ 2.3 indicates highly probable species identification, between 2.0 and 2.3 genus identification and probable species, between 1.7 and 1.99 probable gender, and <1.7 non-reliable identification. The analysis was repeated if the spots resulted in “no peaks found”.

### 4.4. Evaluation and Comparison of Disinfectants Effectiveness: Microbicidal Efficacy Test

To evaluate the antimicrobial effectiveness of the three tested disinfectants, we calculated the inhibition percentage (%) of each one using the following formula:$$\mathrm{Percent}\;\mathrm{of}\;\mathrm{reduction}=100-\frac{\left(TRT*100\right)}{CTR}$$
where TRT = swab 2, 4, 6 (CFU/cm^2^) and CTR = swab 1, 3, 5 (CFU/cm^2^).

## 5. Conclusions

In summary, Argirium-SUNc^®^ formulations show great promise as next-generation biocides due to their unique chemo-physical properties, including different oxidative states of silver (Ag^0^/Ag^n+^), high surface charge, extremely small size (<2 nm), and polygonal shape. Unlike conventional biocides, their antimicrobial activity is based on multiple mechanisms of action, making them less likely to induce microbial resistance. This ensures long-term effectiveness and a broad spectrum of activity against bacteria, biofilms, and fungi. The possibility of having a single compound with both fungicidal and bactericidal activity allows us to carry out future studies aimed at designing mixtures that achieve greater efficiency with minimal economic expenditure. This approach would enhance the reduction in microbial contamination on food contact surfaces without altering any characteristics of foods, thereby significantly impacting food safety and controlling the spread of pathogenic and non-pathogenic microbial agents at a cost accessible even to small food companies.

Other advantages are the low cost and precision of the electrochemical synthesis method. This protocol allows greater control over morphological characteristics such as the size and shape of the produced nanoparticles, ensuring that Argirium SUNc^®^ has high efficacy at a low concentration (1.5 µg/mL). Moreover, this synthesis method using conventional systems and inexpensive reagents [48] does not require the use of stabilizing agents in the reaction in order to obtain a stable dispersion, cutting down additional costs in the production process [49].

Finally, from a green perspective, Argirium SUNc^®^ nanoparticles have a minimal environmental impact, being synthetized using biocompatible ingredients (ultra-pure water) without stabilizing agents or other chemical components.

Thus, in the near future, following additional toxicity studies (not only in vitro but also “in vivo”) to confirm their safety at the concentrations necessary for their biocidal activity and further studies to validate and confirm their antimicrobial properties on the same surfaces as well as on others of different food companies, Argirium SUNc^®^ nanoparticles could be used as innovative antimicrobial substances to produce bio-gels compatible in the medical field and preparations for environments and equipment sanitization, offering a promising alternative or complement to existing sanitizers.

To date, research to better characterize the antibacterial capacity of Argirium SUNc^®^ requires further investigation that includes the properties of the nanomaterial at the atomic level. Additionally, repeated testing on the same surfaces and others will be necessary to evaluate and confirm these results. These investigations are the subject of future investigations that are already underway.

## Figures and Tables

**Figure 1 nanomaterials-15-00295-f001:**
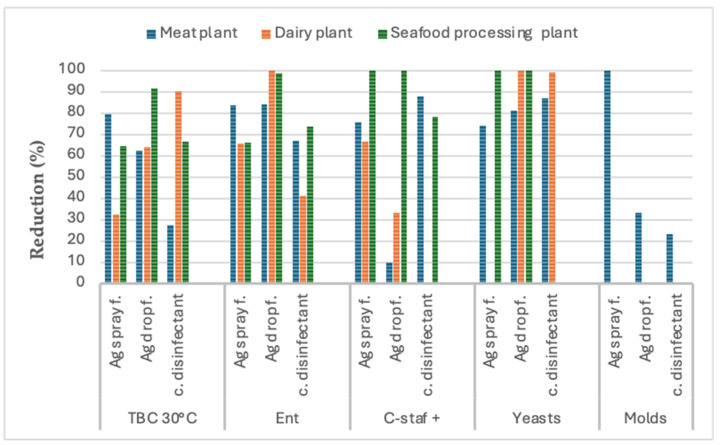
Disinfectant effectiveness on environmental surfaces: graph 1 shows the reduction percentage (%) of each microorganism (total bacterial count, enterobacteria, coagulase-positive staphylococci, yeasts, and molds) after 10 m of exposure to the 3 disinfectants (a conventional chlorine-based disinfectant and Argirium SUNc^®^ in spray and in drop formulation) tested on environmental surfaces (3 for each one) of the 3 different food companies: meat cutting and processing company (blue), dairy plant (orange), and seafood processing plant (green).

**Figure 2 nanomaterials-15-00295-f002:**
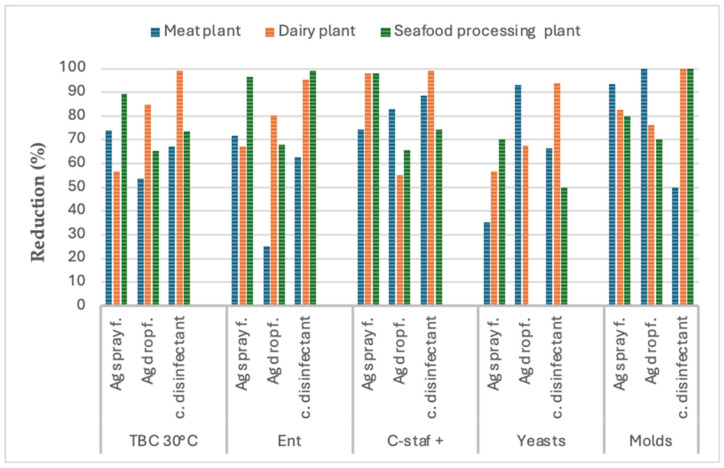
Disinfectant effectiveness on tool surfaces: graph 1 shows the reduction percentage (%) of each microorganism (total bacterial count, enterobacteria, coagulase-positive staphylococci, yeasts, and molds) after 10 m of exposure to the 3 disinfectants (a conventional chlorine-based disinfectant and Argirium SUNc^®^ in spray and in drop formulation) tested on tool surfaces (3 for each one) of the 3 different food companies: meat cutting and processing company (blue), dairy plant (orange), and seafood processing plant (green).

**Figure 3 nanomaterials-15-00295-f003:**
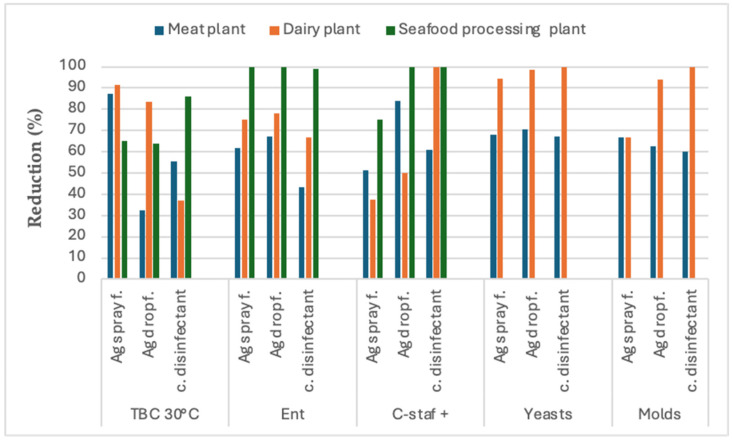
Disinfectant effectiveness on equipment surfaces: graph 1 shows the reduction percentage (%) of each microorganism (total bacterial count, enterobacteria, coagulase-positive staphylococci, yeasts, and molds) after 10 m of exposure to the 3 disinfectants (a conventional chlorine-based disinfectant and Argirium SUNc^®^ in spray and in drop formulation) tested on equipment surfaces (3 for each one) of the 3 different food companies: meat cutting and processing company (blue), dairy plant (orange), and seafood processing plant (green).

## Data Availability

Data are contained within the article.

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
