# Peer review of "Biocidal Properties of New Silver Nanoparticles Argirium SUNc® Against Food Hygiene Indicator Microorganisms"

_nanomaterials, 2025, doi:10.3390/nano15040295_

Round 1

Reviewer 1 Report (New Reviewer)

Comments and Suggestions for Authors

Paper “Biocidal properties of new silver nanoparticles Argirium SUNc® against food hygiene indicator microorganisms” by Mancusi et al. submitted to Nanomaterials

In my opinion, the text is too long, contains repetitive statements and should be shortened. Conclusions should be formulated more briefly.

My question concerns the safety of this preparation for the environment. Does it decompose in the natural environment or should it be destroyed before being released into the environment?

Minor comments:

Please standardize the spelling throughout the text of the abbreviation silver nanoparticles

L 51 Silver nanoparticles (AgNps) or  L 62 AgNPs

L 88 “the bind with sulfur and phosphorous groups of the DNA”       DNA contains phosphorus, but not sulfur

L 270 coagu-lase-positive staphylococci,  coagulase-positive

L 279, L 368  Infact,   should be In fact

L 339-340 indicator microorganisms (coagulase-positive staphylococci, Escherichia coli, enterobacteria, total bacterial count, yeasts and molds)

Please remove (coagulase-positive staphylococci, Escherichia coli, enterobacteria, total bacterial count, yeasts and molds) from the discussion, these microorganisms have already been mentioned several times

L 329, L382, L508    Escherichia coli  please use the abbreviation E. coli

L 508 TXB      please provide the full name of the medium Tryptone Bile X-glucuronide

L 509  violet red bile agar with glucose    change to  Violet Red Bile Glucose Agar

L 551 ARGIRIUM-SUNCs or Argirium SUNc

L 550-558 Conclusions  It should be given in points and made more condensed.

L 587-588 The last sentence is in Italian, was it intent to be so ?

Author Response

Reviewer 1

Comments and Suggestions for Authors

Paper “Biocidal properties of new silver nanoparticles Argirium SUNc® against food hygiene indicator microorganisms” by Mancusi et al. submitted to Nanomaterials

In my opinion, the text is too long, contains repetitive statements and should be shortened. Conclusions should be formulated more briefly.

My question concerns the safety of this preparation for the environment. Does it decompose in the natural environment or should it be destroyed before being released into the environment?

Answer: Thank you for your observation. Argirium SUNc® nanoparticles have a minimal environmental impact, being synthesized using biocompatible ingredients (ultra-pure water) without stabilizing agents or other chemical components. So, they decompose in the natural environment.

Minor comments:

Please standardize the spelling throughout the text of the abbreviation silver nanoparticles

L 51 Silver nanoparticles (AgNps) or  L 62 AgNPs.

L 88 “the bind with sulfur and phosphorous groups of the DNA”       DNA contains phosphorus, but not sulfur

L 270 coagu-lase-positive staphylococci,  coagulase-positive

L 279, L 368  Infact,   should be In fact

L 339-340 indicator microorganisms (coagulase-positive staphylococci, Escherichia coli, enterobacteria, total bacterial count, yeasts and molds)

Please remove (coagulase-positive staphylococci, Escherichia coli, enterobacteria, total bacterial count, yeasts and molds) from the discussion, these microorganisms have already been mentioned several times

L 329, L382, L508    Escherichia coli  please use the abbreviation E. coli

L 508 TXB      please provide the full name of the medium Tryptone Bile X-glucuronide

L 509  violet red bile agar with glucose    change to  Violet Red Bile Glucose Agar

L 551 ARGIRIUM-SUNCs or Argirium SUNc

L 550-558 Conclusions  It should be given in points and made more condensed.

Answer: Thank you for your observation. All the requested corrections have been inserted directly in the manuscript.

L 587-588 The last sentence is in Italian, was it intent to be so ?

Answer: Thank you for your attention, we have corrected the error.

Reviewer 2 Report (New Reviewer)

Comments and Suggestions for Authors

The topic of the article is of great interest. The experiment is well done, but a few aspects need to be clarified.

Minor comments:

L37 "The use of the same bactericides in multiple sectors as medicine" in my opinion, the problem occurs mainly from using the same chemical products for a long time, and bacteria's ability to adapt increases.

L53-54 authors mentioned "numerous studies," even though only one reference exists.

I recommend calculating significant differences between disinfectants effectiveness on environmental surfaces through a statistical analysis. In L226, the authors mentioned "notable variations in the biocidal activity among the different disinfectants." This must be sustained statistically: are there significant differences?

I recommend a clear sentence summarizing the work as a conclusion at the end of the manuscript.

Author Response

Reviewer 2

The topic of the article is of great interest. The experiment is well done, but a few aspects need to be clarified.

Minor comments:

L37 "The use of the same bactericides in multiple sectors as medicine" in my opinion, the problem occurs mainly from using the same chemical products for a long time, and bacteria's ability to adapt increases.

Answer: Thank you for your observation. All the requested corrections have been inserted directly in the manuscript.

L53-54 authors mentioned "numerous studies," even though only one reference exists.

Answer: Thanks for your observation. The term "numerous studies" refers to studies 10, 11, 12, and 13. We have placed the bibliographic reference next to each type of study analyzed.

I recommend calculating significant differences between disinfectants effectiveness on environmental surfaces through a statistical analysis. In L226, the authors mentioned "notable variations in the biocidal activity among the different disinfectants." This must be sustained statistically: are there significant differences?

Answer: Thank you for your observation. We didn’t include this parameter, as the calculated differences were not statistically significant.

I recommend a clear sentence summarizing the work as a conclusion at the end of the manuscript.

Answer: Thank you for your observation. We modified the conclusion of the manuscript.

This manuscript is a resubmission of an earlier submission. The following is a list of the peer review reports and author responses from that submission.

Round 1

Reviewer 1 Report

Comments and Suggestions for Authors

The manuscript is out of the scope of the current journal. The main objective of the study is to investigate the effect of Argirium SUNc® against in two different formulations against the hygiene indicator microorganisms potentially present in three different Italian food industries and compare it with the chemical disinfectant most used by operators for the routine cleaning. There is no information biosynthesis of the nanomaterial used and its characterization and toxicity studies. Hence I feel it is more suitable for a microbiology-related journal.

Comments on the Quality of English Language

Extensive English language editing is required.

Author Response

ow are present in revised manuscript the informations for referee.

In the manuscript:”For this reason, novel silver nanoparticles generated using a reproducible electro- 104 chemical method (Patent EP-18181873) derived from the modify of an old synthesis [30], 105 have been created. These nanoparticles called Argirium Silver Ultra Nano Clusters (Ar- 106 girium SUNc®), are the smallest of all nanoparticles so far studied, with a size < 2 nm 107 [31]. They have a particle structure never observed in a stable form before with a core of 108 metallic Ag0 and the external shells made by Ag+, Ag2+ and Ag3+ silver oxides. These 109 characteristics are responsible for their unique chemo-physical properties. In fact, Ar- 110 girium SUNc® nanoparticles are stable for several months in ultra-pure water solution 111 thanks to a high anionic salvation surrounding (Zpuls value > − 50 mV) [32] and have 112 enhanced redox properties towards biological targets due to the presence of silver oxides 113 on the clusters surface [31]. Therefore, they are considered a novel nanomaterial that 114 have shown their antimicrobial power on either susceptible or resistant strains at a con- 115 centration much lower than that reported for other silver formulations (< 1 μg/mL) [33]. 116 Furthermore, it was seen that Argirium SUNc® were also able to act on bacterial biofilm 117 structures causing its deconstructing (0.625 μg/mL) [34–36]. Regarding their toxicity, a lot 118 of studies present in literature indicate that small nanoparticles present a higher surface 119 area/volume ratio and a stronger bond with the bacterial cell membrane than larger ones 120 responsible for a greater cytotoxicity [37]. However, recent in vitro and in vivo toxicity 121 studies carried out on the Argirium SUNc® nanoparticles against lung epithelial cells, 122 revealed that they are no toxic up at concentration equal to 100 μg/mL [38]. Another 123 study has shown that Argirium SUNc® resulted about 10 times less toxic in human cells 124 than in bacteria [35]. In addition, the same author teams tested them with Galleria 125 mellonella as a pre-clinical toxicity test [35]. From the survival curves of the larvae, they 126 have seen that Argirium SUNc® nanoparticles were no-toxic up to the highest concen- 127 tration possible used 6.8 μg/mL [35]. For this reason, this new type of silver nanoparticles 128 could represent a turning point for the future. Specifically, since resistance to conven- 129 tional biocides used in food industries could be involved in the bigger problem of anti- 130 biotic resistance, Argirium SUNc® nanoparticles could be used as an antimicrobial al- 131 ternative and innovative principle to produce a new generation of bio-gels compatible in 132 the medical field and of preparations for environments and equipment sanitization 133 which don’t favor resistance. 134 In this regard, the aim of the present work was to evaluate the biocidal properties of 135 this novel nanomaterial used as disinfectant in different formulations (spray and drop) 136 and concentrations (1.5, 2, 3 and 4 μg/mL) against the hygiene indicator microorganisms 137 potentially present in a seafood processing plant, a dairy plant and a meat cutting and 138 processing company and to compare their effectiveness with that of the disinfectant most 139 used by operators of the interest food industry.

The nanoparticles are the same batch used previous manuscript published in “Antibiotics” recently.

Reviewer 2 Report

Comments and Suggestions for Authors

The purpose of this study was to evaluate their biocidal properties in two different formulations against the hygiene indicator microorganisms potentially present in three different Italian food industries. The manuscript presents a lot of data analysis and discussion about the germicidal efficacy of Argirium SUNc®. However, all the data in the manuscript lacked standard deviation (data from parallel experiments). In addition, there is a lack of analytical determination of the structure or basic components of Argirium SUNc® in the manuscript, which is significantly related to its bactericidal effect. The following are some suggested modifications that should be double-checked before being considered for publication by Nanomaterials.

Author Response

Statistical accuracy: Reviewer 2 suggested that including data from parallel experiments with standard deviations would improve the robustness of your results.

Now in manuscript are inserted the reply of referee. More information are present for reply and is necessary read the latest manuscript version.

Now are present in figure 1,2,3 the statistical data.

Round 2

Reviewer 1 Report

Comments and Suggestions for Authors

Still, this manuscript is more suitable for a Microbiology related journal than Nanomaterials.

Comments on the Quality of English Language

No comments.

Reviewer 2 Report

Comments and Suggestions for Authors

The author has addressed all my questions. However, the format of some references, such as 41-47, needs to be modified.